# The m⁶A methyltransferase METTL3 regulates muscle maintenance and growth in mice

Jennifer M. Petrosino[1], Scott A. Hinger[1], Volha A. Golubeva[1], Juan M. Barajas[2], Lisa E. Dorn[1], Chitra C. Iyer[3,4], Hui-Lung Sun[5], W. David Arnold[3,4], Chuan He [5] & Federica Accornero [1✉]

Skeletal muscle serves fundamental roles in organismal health. Gene expression fluctuations are critical for muscle homeostasis and the response to environmental insults. Yet, little is known about post-transcriptional mechanisms regulating such fluctuations while impacting muscle proteome. Here we report genome-wide analysis of mRNA methyladenosine (m⁶A) dynamics of skeletal muscle hypertrophic growth following overload-induced stress. We show that increases in METTL3 (the m⁶A enzyme), and concomitantly m⁶A, control skeletal muscle size during hypertrophy; exogenous delivery of METTL3 induces skeletal muscle growth, even without external triggers. We also show that METTL3 represses activin type 2 A receptors (ACVR2A) synthesis, blunting activation of anti-hypertrophic signaling. Notably, myofiber-specific conditional genetic deletion of METTL3 caused spontaneous muscle wasting over time and abrogated overload-induced hypertrophy; a phenotype reverted by co-administration of a myostatin inhibitor. These studies identify a previously unrecognized post-transcriptional mechanism promoting the hypertrophic response of skeletal muscle via control of myostatin signaling.

[1] Department of Physiology and Cell Biology, Dorothy M. Davis Heart and Lung Research Institute, The Ohio State University, Columbus, OH, USA. [2] Department of Pathology, The Ohio State University, Columbus, OH, USA. [3] Division of Neuromuscular Disorders, Department of Neurology, The Ohio State University, Columbus, OH, USA. [4] Department of Physical Medicine and Rehabilitation, The Ohio State University, Columbus, OH, USA. [5] Department of Chemistry, Department of Biochemistry and Molecular Biology, and Institute for Biophysical Dynamics, Howard Hughes Medical Institute, The University of Chicago, Chicago, IL, USA. ✉email: federica.accornero@osumc.edu

The regulation of cellular growth, defined as the accumulation of mass, is one of the most vastly studied but incompletely understood processes. In most systems, cells grow to an optimal size before division. However, muscle is unique from most tissues: continuing to grow in its terminally differentiated state yet unable to undergo hyperplasia[1]. Hence muscle presents a unique opportunity to study cell growth (hypertrophy) uncoupled from cellular proliferation, replication, and division. Muscle comprises nearly 40% of total body mass and is essential for organismal physiological health[2]. With aging, all individuals experience losses of muscle strength and size, and when pathological the loss is termed sarcopenia; a major contributor to loss of physical function, risk of fall-related injuries, and overall mortality in older adults[3,4]. Mechanical overload of muscle in humans through resistance training, or in mice via synergist ablation intervention, is a well-established mean of substantially increasing muscle cell size[5]. However, some individuals lack the ability to increase muscle mass through mechanical overloading and for that reason identifying therapeutic agents capable of increasing and maintaining muscle mass has long been at the center of biomedical focus.

Two key pathways have been identified as upstream regulators of muscle size and homeostasis. The first pathway, known as IGF/PI3K/AKT/mTORC1, is anabolic and responsible for the positive regulation of protein synthesis. The second and opposing pathway, i.e. myostatin activation of activin receptors and downstream SMAD signaling, is catabolic and responsible for driving atrophic signaling[1]. In the 1960s, Bullough first introduced the idea that negative regulators of growth, known at the time as chalones, were key determinants of tissue size. In 1997, myostatin, a member of the transforming growth factor-beta (TGF-beta) superfamily, also known as growth/differentiation factor-8 (GDF-8), was identified as the first chalone-type molecule[6], and today is one of the best-known negative regulators of skeletal muscle growth[7]. Many diseases that involve muscle atrophy feature impairments in signaling in the myostatin pathway[8].

The identification of chemical modifications to the cell's transcriptome, and the functionally relevant changes that these modifications elicit, has birthed the field of epitranscriptomics. Much interest has focused on N6-methyladenosine (m6A); the most conserved, abundant, internal mRNA modification[9]. The co- and post-transcriptional reversible transfer of a methyl group to the N6 position of adenosines in mRNA is mediated by a multi-protein complex in which methyltransferase-like-3 (METTL3) bears the enzymatic activity[10,11]. Hence, a paradigm exists where m6A dynamically changes as a post-transcriptional means of regulating and effecting cellular adaption in response to specific stimuli. How this process influences skeletal muscle homeostasis is not currently known.

Here, we identified the METTL3-m6A axis as a post-transcriptional regulator of the myostatin pathway that is critical for the control of muscle size and growth. Using unbiased transcriptomic mapping of m6A dynamics in response to muscle-overload induced hypertrophy, as well as genome-wide mapping of muscle ribosome-associated transcripts to determine METTL3-dependent regulation of mRNA translation, we determined activin receptors as direct targets of post-transcriptional regulation in muscle. Further investigation into the role of METTL3 in the modulation of muscle homeostasis, through gain- and loss-of-function mouse models, revealed that METTL3 is required for muscle size maintenance and hypertrophic responses. Our findings describe a post-transcriptional mechanism acting on the myostatin pathway and consequently illuminate the critical role of METTL3 and m6A in controlling skeletal muscle size.

## Results

**m6A increase and remodeling accompanies skeletal muscle hypertrophy**. To determine the contribution of m6A to skeletal muscle hypertrophy, we first assessed the level of this modification on RNA from mice subjected to mechanical overload (Fig. 1a, b) and found a global increase of m6A levels in overloaded muscles (Fig. 1b). To understand the origin of the observed m6A increase, we then tested the levels of METTL3, the enzyme responsible for m6A formation in mRNA. METTL3 expression was significantly elevated during overload (Fig. 1c, d), suggesting an overall potentiation of the METTL3-m6A axis during adult hypertrophic muscle growth. To gain insights into the identity of the m6A-modified mRNAs at baseline and following muscle overload, we performed m6A RNA immunoprecipitations followed by sequencing (meRIP-seq)(Fig. 1e). While a core set of m6A peaks on the transcriptome were unaffected by the hypertrophic stimulus, our analysis revealed remodeling of the m6A landscape in the overloaded muscle on 1674 mRNAs with 1704 m6A peaks gained on 870 new transcripts and 1591 m6A peaks lost by 804 mRNAs following adaptation to overload (Fig. 1f, g and Supplementary Data 1). Analysis of m6A location on transcripts also revealed enrichment of this modification around mRNA stop codons with subtle differences in density and frequency between baseline and overload conditions (Fig. 1h, i). Gene ontology analysis of the m6A-modified muscle transcriptome further highlighted the stable presence of this modification of mRNAs coding for transcription regulators, while the top enriched category of overload-responsive transcripts was represented by those coding for proteins involved in the regulation of phosphorylation-dependent cell signaling (Fig. 1j, k). These data hint at crosstalk between m6A-dependent post-transcriptional gene regulation, transcription, and post-translational pathways. Altogether, these results demonstrate that increased METTL3 expression and m6A abundance mark the hypertrophic response of skeletal muscle and that the m6A-targeted transcriptome remodels during overload-induced hypertrophy to affect specific enzymatic muscle processes.

**METTL3 is essential for adaptation to hypertrophic muscle stress**. To determine the role of METTL3 in adult skeletal muscle and the consequences of manipulating its levels, we generated a myofiber-specific METTL3 loss-of-function mouse model where deletion of this gene is induced by a tamoxifen-inducible Cre under the control of human skeletal actin (HSA) promoter (METTL3 muscle knock out, M3-mKO)(Fig. 2a). Littermate mice containing solely the loxP sites or Cre were used as controls for wild-type expression of METTL3 (WT) after being subjected to equal tamoxifen administration as their M3-mKO siblings. The efficacy of this model to reduce METTL3 expression was confirmed at both the transcript and protein level using total muscle extracts 14 days post tamoxifen administration (Fig. 2b, c). This short-term deletion strategy did not result in changes in body, heart or skeletal muscle weights (Fig. 2d–h). To determine if METTL3 is necessary for adult skeletal muscle hypertrophy we then subjected M3-mKO mice and littermate controls to muscle overload. Following 14 days of overload, the plantaris muscle from M3-mKO mice failed to increase in weight to the same extent as the WT controls (Fig. 2i). Muscle cell-size analysis further revealed abrogation of overload-induced increase in myofiber cross-sectional in M3-mKO mice (Fig. 2j, k). Thus, METTL3 is essential for muscle cell growth during the hypertrophic response of skeletal muscle to mechanical overload.

**Enhancing the METTL3 pathway favors skeletal muscle growth**. Demonstrating the necessity of METTL3 for hypertrophy opened the question as to whether increasing METTL3 expression in skeletal muscle is sufficient to potentiate the

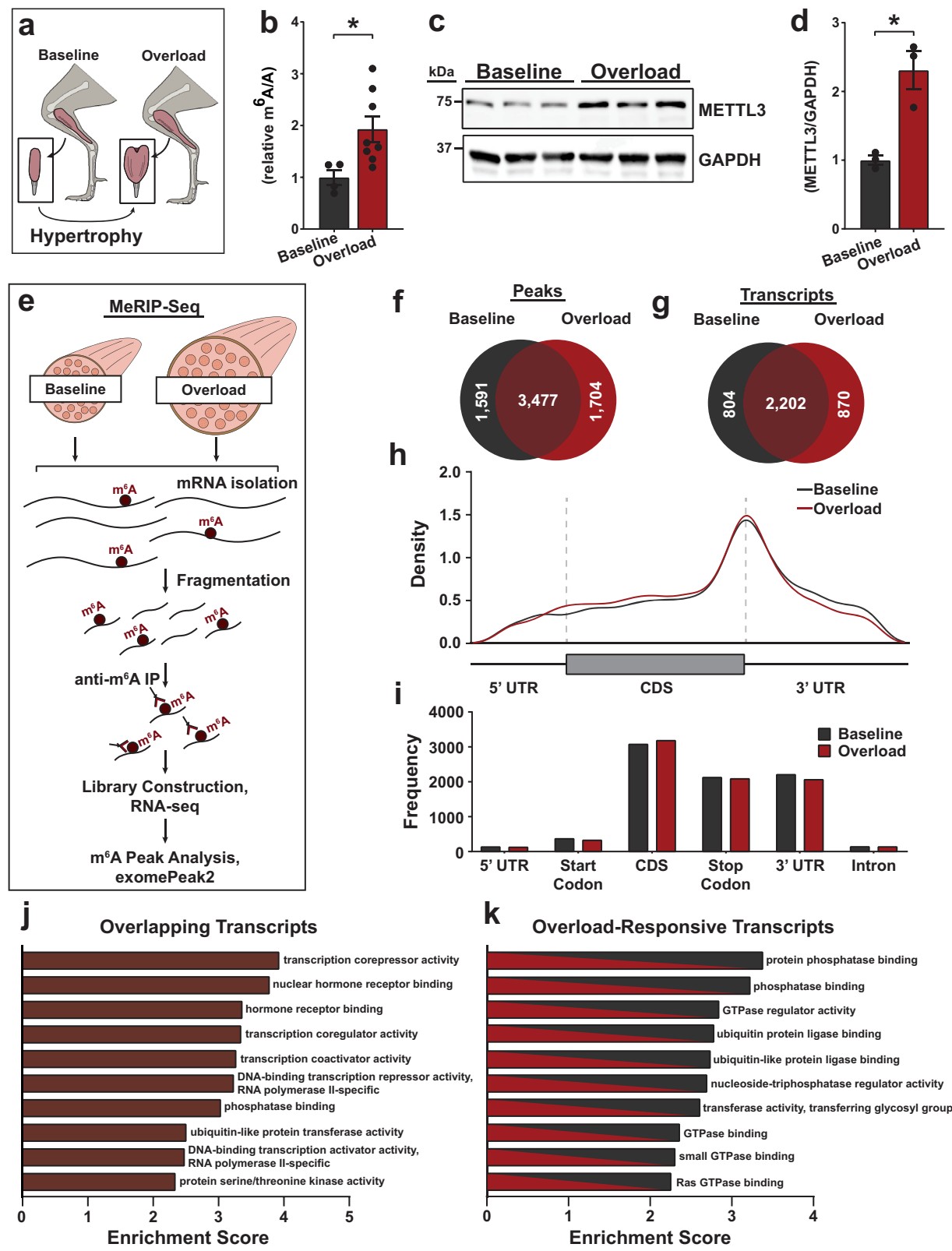

hypertrophic response to muscle overload. For this, a plasmid encoding for Myc-tagged human METTL3 (M3-OE), or Myc-tagged empty vector controls (Ctrl), were electroporated into the plantaris of mice undergoing overload induction to examine the effect of METTL3 overexpression on muscle growth (Fig. 3a). The presence of the exogenous METTL3 construct was confirmed 14 days following the procedure (Fig. 3b) and its functionality was validated by the resulting increase in m6A content (Fig. 3c). Strikingly, METTL3 overexpressing muscles had an enhanced hypertrophic response to overload evidenced by increase plantaris weight (Fig. 3d) and myofiber cross-sectional areas (Fig. 3e, f). We then tested if inundating the METTL3 pathway is sufficient to potentiate skeletal muscle growth during postnatal development. For this, we systemically injected adeno-associated viruses (AAV)

**Fig. 1 METTL3 and m⁶A remodels with muscle growth. a** Schematic depicting muscle overload-induced hypertrophy of the plantaris muscle created in Adobe Illustrator. **b** Quantification of m⁶A level relative to total adenosine (m⁶A/A) as determined by ELISA in wild-type baseline and overloaded muscles. **c** Western blot and **d** quantification of METTL3 expression normalized to GAPDH in baseline and overloaded muscles. **e** Schematic of workflow of MeRIP-Seq experiment created in Adobe Illustrator. **f** Venn diagram of peaks enriched (FC > 1.5) in baseline and overloaded plantaris samples. **g** Venn diagram of transcripts with m⁶A peaks that were found to be m⁶A-targeted in baseline, overloaded, or common between both. **h** Peaks detected in baseline and overloaded were plotted via density across mRNA regions (including 5′ UTR, start codon, coding sequence (CDS), stop codon, and 3′ UTR). UTR untranslated region. **i** Peaks detected in baseline and overloaded samples were plotted via frequency across the indicated mRNA regions. **j** Gene ontology (GO) analysis of overlapping transcripts or **k** overload-responsive transcripts enriched with m⁶A peaks in either baseline and overloaded samples. The enrichment score is based on a reference database of protein-coding genes, and all GO categories plotted were found to have a false discovery rate (FDR) < 0.05. Biological animal replicates: $n = 3$ per group in panel **d** and **f–k**; $n = 4$ (baseline) and 8 (overload) in panel **b**. Data are presented as the mean ± SEM. *$P < 0.05$, by two-sided Student's $t$ test for comparisons between baseline and overloaded muscles. Source data are provided as a Source Data file.

encoding for mouse METTL3 or control viral vectors into post-natal day 3 mice (Fig. 3g). Assessment of METTL3 over-expression 8 weeks post-AAV delivery showed efficient overexpression in the plantaris muscle (Fig. 3h). The higher METTL3 content was accompanied by a spontaneous increase in plantaris muscle weight (Fig. 3i) and this METTL3-driven hypertrophic response was confirmed at the myofiber level (Fig. 3j and Supplementary Fig. 1a). While different muscle types can have different propensity to undergo hypertrophy, systemic delivery of AAV-METTL3 increased myofiber cross-sectional area in all the tested muscles, suggesting a positive general effect of METTL3 on muscle mass regulation (Supplementary Fig. 1b–d). Finally, we tested if METTL3 can drive a spontaneous increase in muscle mass even in the absence of hypertrophic triggers. For this we injected AAV METTL3 or control viral vectors into the tibialis anterior muscle of 5 months old mice (Fig. 3k). Assessment of METTL3 overexpression 8 weeks post-AAV delivery showed efficient overexpression in the tibialis anterior muscle (Fig. 3l). The higher METTL3 content was accompanied by a trending increase in tibialis anterior muscle weight (Fig. 3m) and a significant increase in its myofiber size (Fig. 3n). Thus, METTL3 is pro-hypertrophic in skeletal muscle.

**METTL3 is necessary for skeletal muscle maintenance and function.** Maintenance of a healthy muscle mass is essential to prevent morbidity and mortality in the elderly population, during forced inactivity, and during chronic diseases such as heart failure and cancer[2,12,13]. To assess if dysregulation of the METTL3 pathway is sufficient to drive muscle wasting, we performed a chronic deletion experiment where METTL3 was knocked out from adult muscle for 8 months (Fig. 4a). While not significantly impacting body or heart weights (Fig. 4b, c), chronic long-term deletion of METTL3 led to a decrease in the mass of all skeletal muscles analyzed (Fig. 4d–h). The observed muscle wasting was confirmed at the cellular level by quantifying the cross-sectional myofiber area of tibialis anterior, plantaris, and soleus muscles (Fig. 4i–l). The muscle phenotype driven by loss of METTL3 in myofibers is progressive, as 6 months of deletion showed a less severe phenotype (Supplementary Fig. 2a–h) and no significant wasting was observed after only 5 weeks of deletion (Supplementary Fig. 2i–p). To determine functional consequences of the muscle wasting observed after 8 months of METTL3 deletion, we performed in vivo muscle torque measurements, which revealed a decrease in plantar-flexion tetanic muscle force in METTL3-deficient mice (Fig. 4m). M3-mKO mice further showed defective running performance and lower maximum oxygen consumption on metabolic treadmill testing (Fig. 4n, o). Altogether, these data demonstrate that METTL3 is necessary for maintaining muscle mass and function over time.

**METTL3-dependent regulation of muscle m⁶A-mRNA translation governs hypertrophic gene regulation.** The formation of m⁶A on mRNA can post-transcriptionally impact gene expression by altering the ability of modified mRNAs to be translated. To mechanistically understand how METTL3 modulates muscle size we crossed M3-mKO mice and WT controls to a mouse model containing an HA-tag on ribosomal protein 22 (RPL22) under the control of a skeletal actin promoter, thus allowing for isolation of myofiber-specific ribosomes (Fig. 5a). Expression of HA-RPL22 was confirmed (Fig. 5b) and ribosome-associated transcripts were analyzed by immunoprecipitations of myofiber ribosomes from total muscle extracts using HA antibodies, followed by sequencing of 'pulled down' mRNA as a proxy for the myofiber-specific 'translatome' as compared to total normalizing mRNA inputs (Fig. 5c). Bioinformatic cross-analysis of ribosome-associated mRNAs, from baseline muscles with or without METTL3, against all the baseline m⁶A-containing mRNAs previously identified by meRIP-seq revealed that ~ 39% of the mRNAs undergoing METTL3-dependent differential translation were also m⁶A-containing and therefore direct targets of the METTL3 pathway (Fig. 5d). In particular, out of 557 differentially translated transcripts (453 enriched in WT, plus 104 enriched in mKO), 216 were m⁶A containing (178 in the WT enrichment, plus 38 in the mKO enrichment) (Fig. 5d and Supplementary Data 2). Gene ontology analysis of these overlapping transcripts revealed the TGF-beta superfamily receptor category as the most enriched (Fig. 5e). Within this category of genes, activin type 2 A receptor (*Acvr2a*) was an exemplary transcript with increased translation in muscles from METTL3 KOs (Fig. 5f). Since activin receptors are regulators of muscle mass, we focused on the behavior of these transcripts. First we confirmed that the m⁶A modification on *Acvr2a* is METTL3 dependent (Fig. 5g), and that loss of METTL3 in muscle leads to increased translation of *Acvr2a* mRNA (Fig. 5h) and synthesis of ACVR2A protein (Fig. 5i). RNA binding proteins of the YTH domain-containing family (YTHDF) are key regulators of m⁶A-modified transcripts[14]. YTHDF2 was specific in binding *Acvr2a* mRNA (Fig. 5j). YTHDF2 favors the decay of interacting transcripts[15], suggesting METTL3-dependent m⁶A methylation could affect *Acvr2a* stability. Indeed, METTL3 decreases the stability of *Acvr2a* mRNA (Fig. 5k). While *Acvr2a* is post-transcriptionally regulated by METTL3 in muscle, another member of the activin receptor family, *Acvr2b*, is downregulated during muscle overload (Supplementary Fig. 3a), independent of METTL3 and m⁶A (Supplementary Fig. 3b, c). m⁶A modification of *Acvr2a* mRNA further increases during muscle overload (Fig. 6a), suggesting potentiation of METTL3 effects on this pathway during stress adaptation. Activin receptor activation exerts anti-hypertrophic effects through intracellular SMAD3 phosphorylation. Therefore, we probed for this post-translational modification and found that METTL3-deficient mice have aberrant SMAD3 phosphorylation

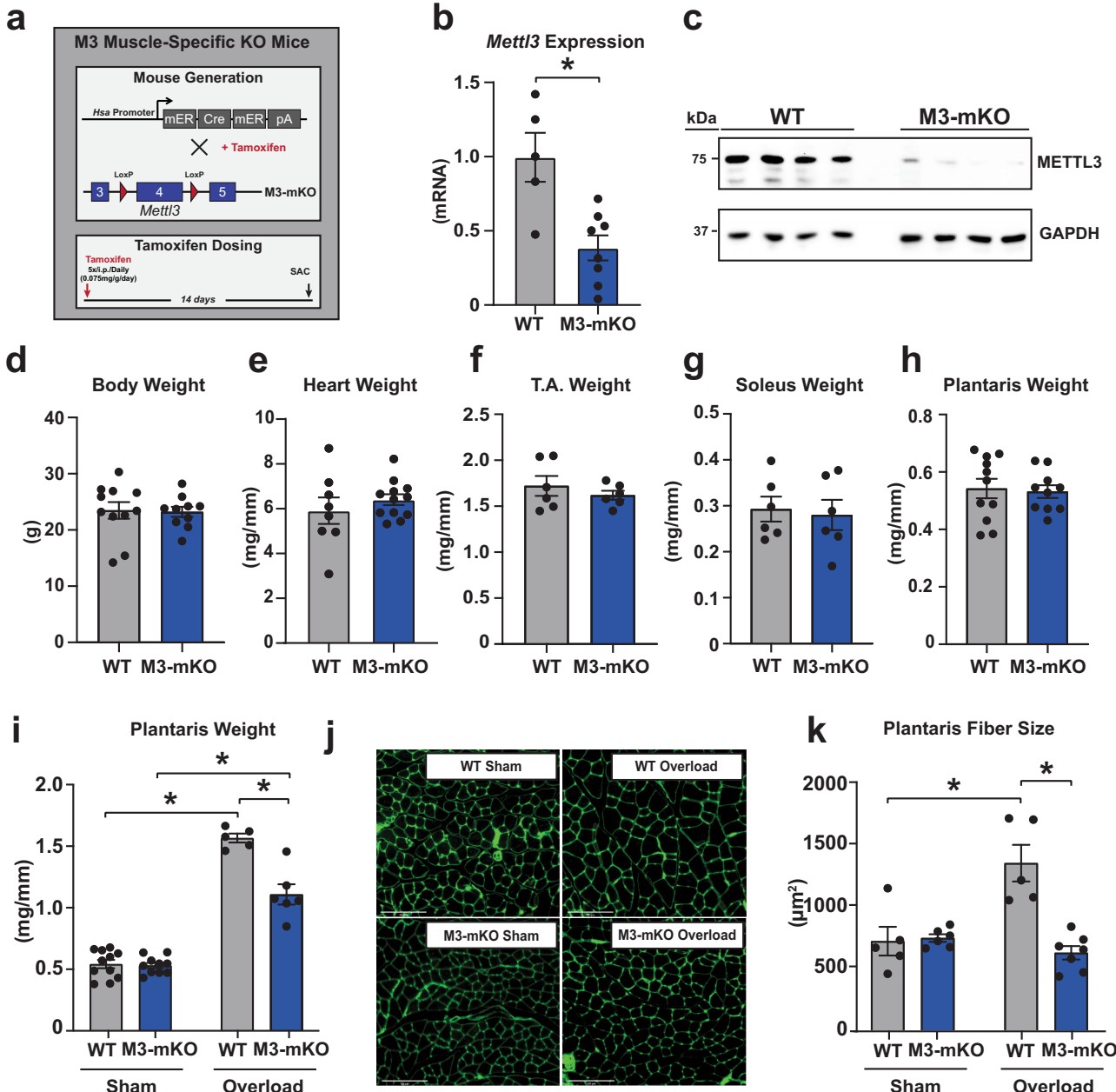

**Fig. 2 METTL3 is essential for the growth response of muscle. a** Schematic of muscle-specific METTL3 knock out (M3-mKO) mouse generation. **b** qPCR analysis of *Mettl3* mRNA expression in the plantaris muscles of WT and M3-mKO mice. **c** Western blot of METTL3 protein expression and GAPDH loading control in WT and M3-mKO muscles. **d** Body weight, **e** heart weight, **f** tibialis anterior (T.A.) weight, **g** soleus weight, and **h** plantaris weight in WT and M3-mKO mice 2 weeks post tamoxifen injections to induced muscle-specific deletion of *Mettl3*. Tibia length (TL) was used to normalize cardiac and skeletal muscle weights. **i** Plantaris weight, **j** representative wheat germ agglutinin (WGA; green) stained images and **k** plantaris fiber size at baseline or 14 days after synergist ablation surgery in WT and M3-mKO mice. Biological animal replicates: $n = 5$ (WT) and 8 (mKO) in panel b; $n = 11$ (WT) and 10 (mKO) in panel **d** and **h**; $n = 8$ (WT) and 12 (mKO) in panel **e**; $n = 6$ (WT) and 6 (mKO) in panel **f** and **g**; $n = 11$ (WT sham), 10 (mKO sham), 5 (WT overload), and 6 (mKO overload) in panel **i**; $n = 5$ (WT sham), 6 (mKO sham), 5 (WT overload), and 7 (mKO overload) in panel **k**. Data are presented as the mean ± SEM. *$P < 0.05$, by 2-sided Student's *t* test for comparisons between WT and M3-mKO mice, or by 2-way ANOVA with Tukey's HSD multiple-comparison test for comparison of the means of WT and M3-mKO mice and baseline and during overload. Scale bar = 125 µm. Source data are provided as a Source Data file.

in overloaded muscles (Fig. 6b). In this condition, a modest negative effect on AKT phosphorylation was also observed (Supplementary Fig. 4a), while an additional growth-modulating pathway, forkhead box O-3 (FOXO3), showed more variability (Supplementary Fig. 4b). Anti-hypertrophic SMAD3 transcriptional targets, such as *Murf1* (muscle-specific ring finger protein 1) and *Mafbx* (muscle-atrophy F-box protein) were increased in METTL3-deficient muscles (Fig. 6c, d). Thus, formation of m6A

by METTL3 affects TGF-beta superfamily signaling through regulation of activin receptor mRNA translation.

**Inhibition of the activin receptor pathway rescues METTL3 KO muscle phenotypes.** The impact of METTL3-mediated m6A modifications on the TGF-beta superfamily receptor highlighted the role of METTL3 as a post-transcriptional regulator

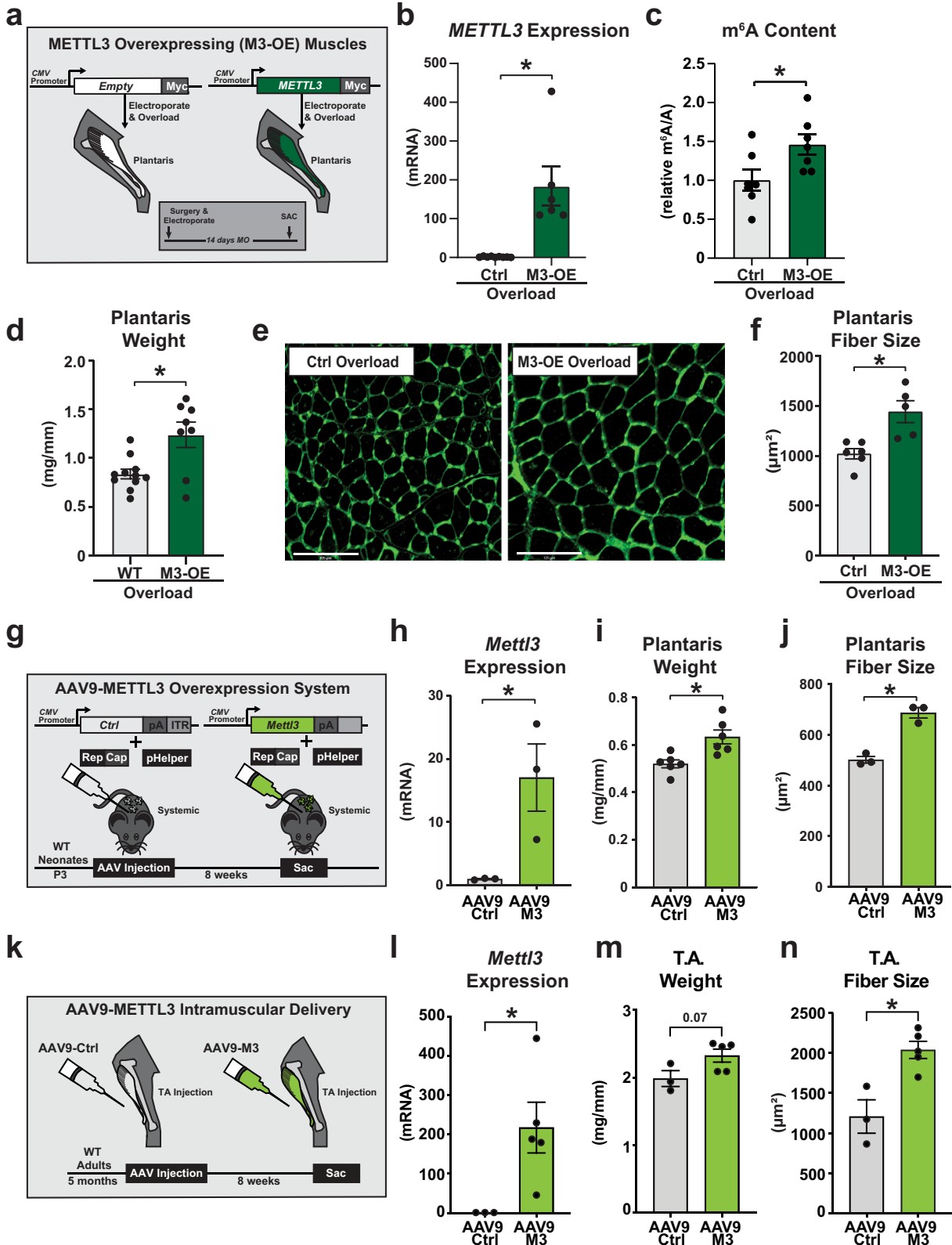

of the myostatin pathway. To demonstrate the causal relationship between activin receptor biology and METTL3-dependent regulation of muscle mass we administered ACE-031, an inhibitor of activin receptor ligands (namely myostatin as this protein is a predominant ligand for these receptors in skeletal muscle) (Fig. 6e). Co-administration of the myostatin inhibitor fully rescued the impaired hypertrophic growth

observed in M3-mKO mice following 14 days of muscle overload as evaluated at the organ level by plantaris weight and at the myofiber level by the cross-sectional area (Fig. 6f–h). Altogether, our data demonstrate that METTL3 regulates muscle growth by fine-tuning activin receptor translation affecting adult skeletal muscle maintenance, hypertrophic growth, and function (Fig. 6i).

**Fig. 3 METTL3 is pro-hypertrophic in skeletal muscle. a** Schematic of overexpression of Myc-control or Myc-tagged METTL3 plasmid through DNA electroporation in overloaded WT muscles created in Adobe Illustrator. **b** qPCR analysis of human *METTL3* expression in the plantaris muscles of Myc-control (Ctrl) or Myc-tagged METTL3 (M3-OE). **c** Quantification of $m^6A$ level relative to total adenosine ($m^6A/A$) as determined by ELISA in electroporated muscles. **d** Plantaris weight, **e** representative wheat germ agglutinin (WGA; green) stained images, and **f** plantaris fiber size 14 days after synergist ablation surgery in control (Ctrl) or Myc-tagged METTL3 (M3-OE) plantaris muscles. **g** Schematic of overexpression through AAV9-Ctrl or AAV9-METTL3 injections into neonatal WT mice and experimental endpoint created in Adobe Illustrator. **h** qPCR analysis of *Mettl3* expression in the muscles of AAV9-Ctrl or AAV9-M3 muscles. **i** Plantaris weight, and **j** plantaris fiber size 8 weeks after AAV injection in AAV9-Ctrl or AAV9-M3 animals. **k** Schematic of overexpression through AAV9-Control (AAV9 Ctrl) or AAV9-METTL3 (AAV9 M3) intramuscular injections into the tibialis anterior (T.A.) of 5-month-old WT mice and experimental endpoint created in Adobe Illustrator. **l** qPCR analysis of *Mettl3* expression, **m** T.A. weight, and **n** T.A. myofiber size of AAV9 Ctrl or AAV9 M3 mice following 8 weeks of injection. Biological animal replicates: $n = 7$ (Ctrl) and 6 (M3-OE) in panel **b**; $n = 7$ (Ctrl) and 7 (M3-OE) in panel **c**; $n = 11$ (Ctrl) and 8 (M3-OE) in panel **d**; $n = 6$ (Ctrl) and 5 (M3-OE) in panel **f**; $n = 3$ per group in panel **h**, **j**, and **n**; $n = 6$ per group in panel **i**; $n = 4$ (Ctrl) and 5 (AAV9-M3) in panel **l**; and $n = 3$ (Ctrl) and 4 (AAV9-M3) in panel **m**. Data are presented as the mean ± SEM. $*P < 0.05$, by two-sided Student's $t$ test for comparisons between 2 groups. Scale bar $= 125\,\mu m$. Source data are provided as a Source Data file.

## Discussion

The myriad of complex mechanisms that operate at the mRNA level to control gene expression makes it particularly difficult to fully understand how cells sense and respond to their environment, especially in the complex physiological systems of higher vertebrates. In this regard, how any naturally occurring chemical RNA modification could play a central role in skeletal muscle physiology has not been investigated prior to this study. The work presented shows how differential methylation of adenosines on defined sets of mRNAs marks and controls the regulation of muscle cell size, function, and adaptation to hypertrophic stress.

The discovery that $m^6A$ is enriched on mRNAs encoding for components of the universally critical TGF-beta superfamily pathway highlights the significance of our findings even beyond muscle. The TGF-beta pathway components are conserved, ubiquitous and pleiotropic regulators of cell behavior in metazoans[16,17]. This superfamily comprises TGF-beta isoforms, activins, inhibitins, bone morphogenetic proteins, and growth/differentiation factors such as myostatin[18]. Cells transduce the signaling exerted by these proteins predominantly though TGF-beta and activin receptors[19,20]. An intersection between $m^6A$ and the TGF-beta signaling was observed in human stem cells by the discovery of a role for SMAD2/3 transcription factors in mediating METTL3 target selectivity[21]. Our study highlights the mRNAs coding for activin receptors as particularly important $m^6A$-targets undergoing post-transcriptional regulation. Activin receptor activation is the single most potent negative regulator of cell size in skeletal muscle[22].

METTL3-mediated methylation either increases or decreases stability and translation of its $m^6A$-modified mRNA targets, likely depending on which $m^6A$-binding protein engages the modified transcript[23,24]. Here, METTL3 led to a decrease in the stability of the activin type 2A receptor. This was likely mediated by YTHDF2, as this specific $m^6A$-RNA binding protein interacted with activin type 2A receptor transcript. This suggests that in this particular system, $m^6A$ acts as a repressor and raises the possibility that in this context YTHDF2 may play a major role in its regulation. At least, within the cytoplasmic $m^6A$-binding proteins, YTHDF2 is known to drive mRNA degradation[15,25]. Future work will be needed to pinpoint the mechanisms underpinning the translational fate of specific $m^6A$-modified muscle mRNAs through the activity of RNA-binding proteins. While activin type 2 A receptor was critically regulated at the post-transcriptional level by the METTL3-$m^6A$ pathway, we cannot currently exclude a potential contribution of additional pathways to this regulation. Another member of activin signaling, activin type 2B receptor, showed a drop in total transcript level following muscle overload in the absence of a clear role for METTL3 in this process. This suggests diversification on how gene regulation evolved in muscle to modulate activin receptor signaling under growth stimuli,

likely providing a protective regulatory multilayer of pathway potency tuning.

Our study is the first to address the role of the METTL3-$m^6A$ pathway in skeletal muscle homeostasis and hypertrophy. By utilizing gain- and loss-of-function METTL3 animal models, we profiled in vivo $m^6A$ dynamics during adult muscle stress responses and defined the direct consequences of this modification to mRNA translation through unbiased approaches. While manipulation of METTL3 level in the context of growing muscle was quick and potent in regulating organ size, loss of METTL3 also led to a progressive wasting phenotype in the absence of other stressors. The fact that several months are necessary to reach a significant effect at baseline is likely dependent upon the availability of TGF superfamily ligands that would be necessary to accentuate the role of METTL3 in muscle. Six studies exist that have touched on the importance of $m^6A$ in myogenic cells[26–31]; none have defined the impact of manipulating this pathway for skeletal muscle development or adult physiology. The findings described here reveal a previously unrecognized control point for activin receptor modulation via the in vivo necessity of $m^6A$ for the maintenance of muscle mass and function, including its essentiality for the induction of skeletal muscle hypertrophy. While the field of epitranscriptomics in skeletal muscle is in its infancy, this current study highlights the exciting potential of manipulating $m^6A$ levels to mitigate age-driven muscle dysfunction, with obvious therapeutic ramifications to counteract the deleterious consequences of skeletal muscle wasting.

## Methods

**Ethics declarations**. All presented experiments comply with the standards set forth by the Institutional Animal Care and Use Committee at The Ohio State University, and the Guide and Care and Use of Laboratory Animals published by the US National Institute of Health. All protocols are approved by The Ohio State University Institutional Animal Care and Use Committee and Institutional Biosafety Committee.

**Animal generation**. Male and female C57BL6/J mice up to 14 months of age were used in this study. Mice were housed at 72° Fahrenheit under a 12-h light/12-h dark cycle and maintained on a standard chow diet. Mice had ad libitum access to food and water. The generation of *Mettl3* LoxP-targeted (flox; fl) mice (*Mettl3*[fl/fl]) was previously described[32]. *Mettl3*[fl/fl] mice were crossed with mice expressing tamoxifen-inducible Cre recombinase gene under the control of the skeletal myofiber-specific human skeletal alpha actin (HSA) promoter[33], i.e. HSA-MerCreMer mice, to obtain myofiber-restricted deletion of *Mettl3* (M3-mKO) in adult mice. Mice that were wild-type for *Mettl3*, but expressing Cre recombinase or the flox allele without Cre were used as controls. To generate mice with myofiber-specific HA-tagged ribosomes, M3-mKO or HSA expressing controls were crossed with homozygous RiboTag[34] (RPL22[HA], #011029, The Jackson Laboratory) mice. Tamoxifen (MilliporeSigma, Burlington, MA, USA) was mixed in sesame oil at 25 mg/ml. Mice were given doses of tamoxifen at 0.075 mg/g/d for 5 consecutive days by intraperitoneal injection. Animals utilized for studies examining the chronic impact of *Mettl3* deletion were subjected to doses of tamoxifen at 0.075 mg/g/d for 5 consecutive days by intraperitoneal injection once every 6 weeks till experiment termination. Maintenance of *Mettl3* deletion using a tamoxifen

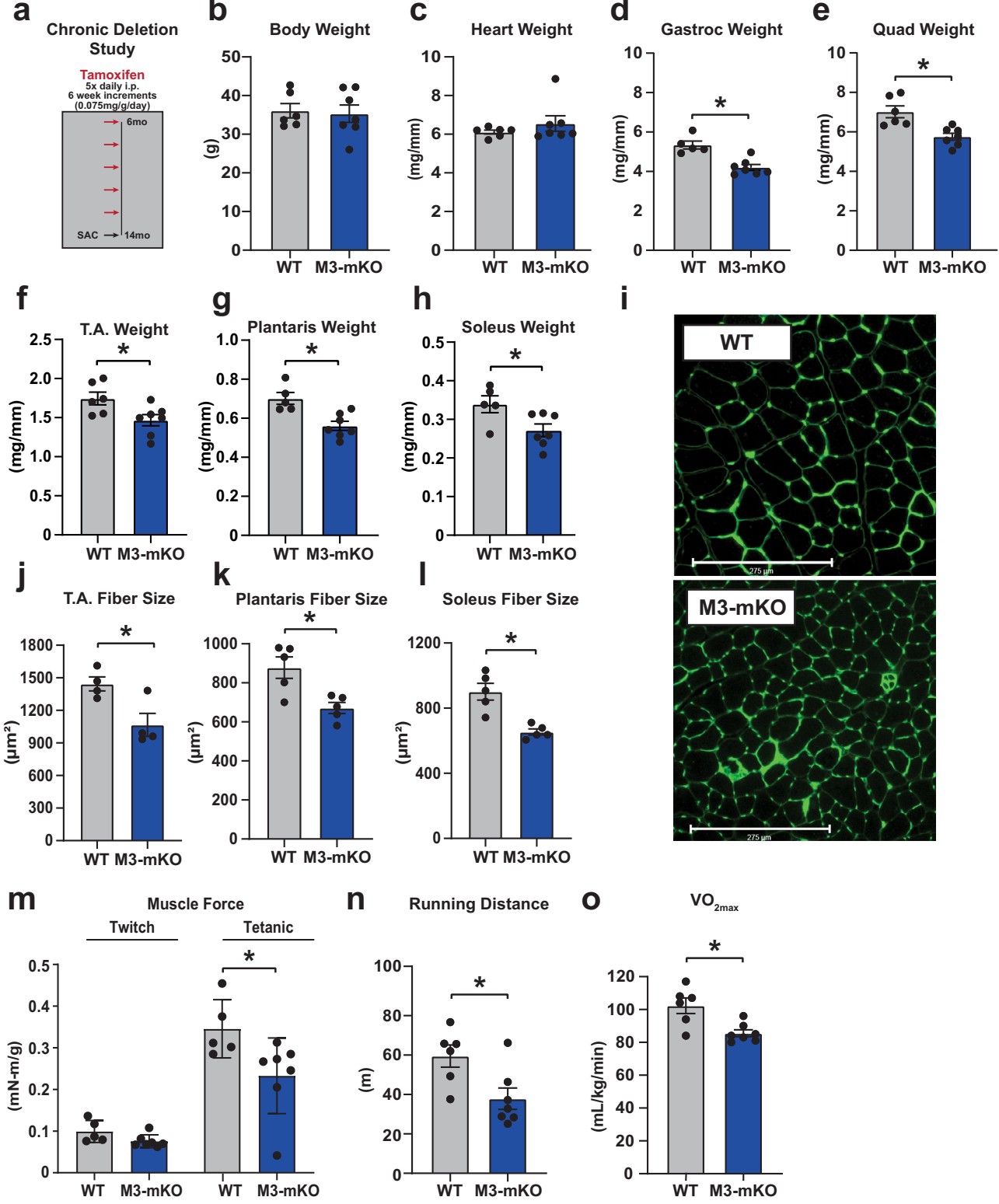

**Fig. 4 Chronic deletion of METTL3 drives muscle atrophy. a** Schematic of M3-mKO chronic deletion study. **b** Body weight, **c** heart weight, **d** gastrocnemius (gastroc) weight, **e** quadriceps (quad) weight, **f** tibialis anterioris (T.A.) weight, **g** plantaris weight, and **h** soleus weight of WT and M3-mKO mice. Tibia length (TL) was used to normalize cardiac and skeletal muscle weights. **i** Representative wheat germ agglutinin (WGA; green) stained images of WT and M3-mKO mice at 14 months of age. **j** T.A., **k** plantaris, and **l** soleus fiber size at in WT and M3-mKO mice following chronic deletion. **m** In vivo muscle twitch and tetanic torque measurements in 14 months old WT and M3-mKO mice. **n** Maximal running distance and **o** maximal running oxygen consumption following a graded maximal exercise test in WT and M3-mKO mice at 14 months. Biological animal replicates: $n = 6$ (WT) and 7 (mKO) in panel **b**, **c**, **e**, **f**, **n**, and **o**; $n = 5$ (WT) and 7 (mKO) in panel **d**, **g**, **h**, and **m**; $n = 4$ (WT) and 4 (mKO) in panel **j**; $n = 5$ (WT) and 5 (mKO) in panel **k** and **l**. Data are presented as the mean ± SEM. *$P < 0.05$, by two-sided Student's $t$ test for comparisons between WT and M3-mKO mice. Scale bar = 275 μm. Source data are provided as a Source Data file.

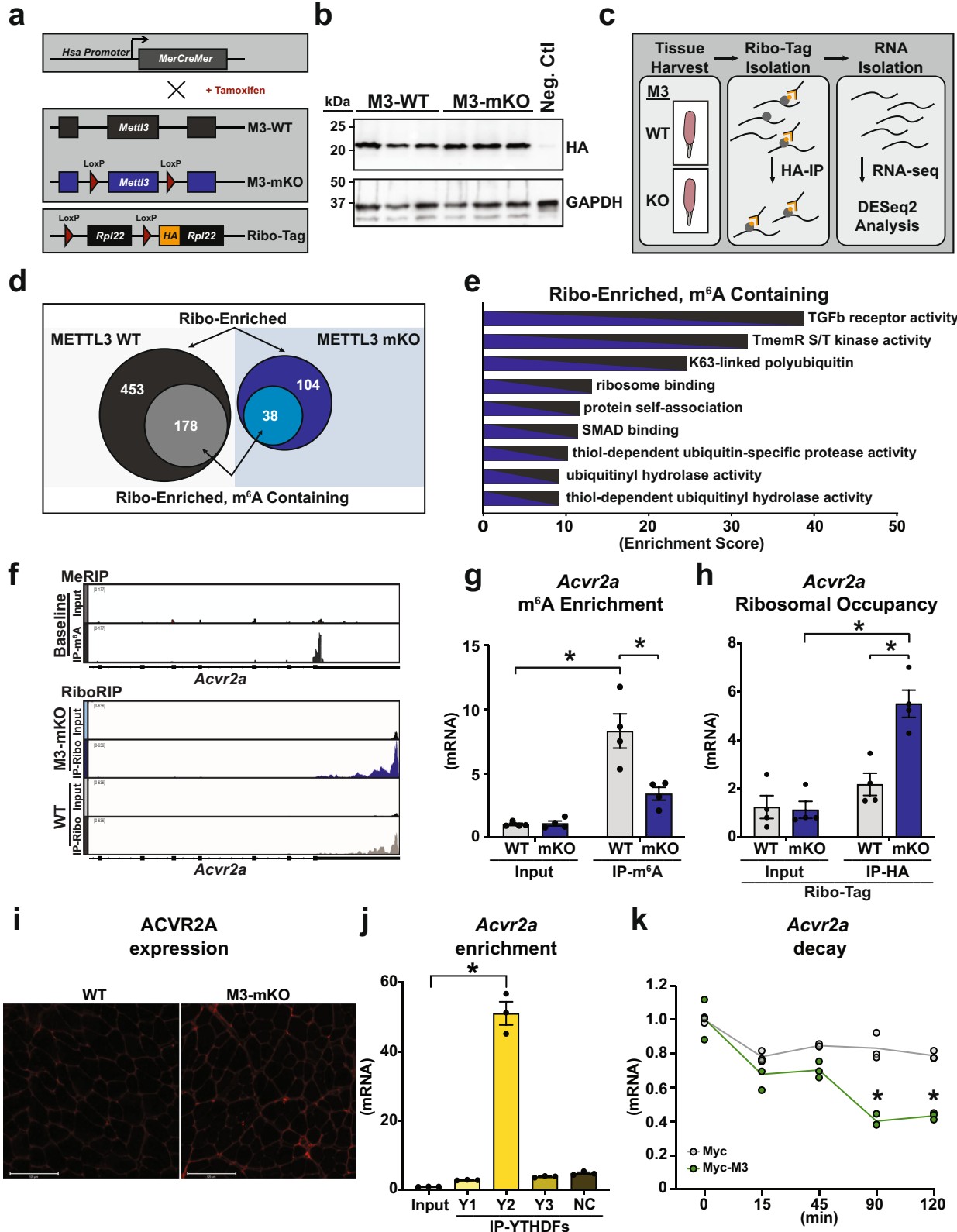

chow regimen (40 mg/kg/day, ENVIGO, TD.130860) was used in muscle-overloaded mice.

**Animal procedures and treatments**. With the exception of AAV injections in postnatal day 3 mice, all experimental procedures were initiated on mice between 3 and 6 months of age.

Hypertrophy stimulating plantaris overload was achieved through bilateral synergistic ablation of the soleus and gastrocnemius. In short, lateral incisions were

made on the bilateral lower hindlimbs, allowing for the exposure and subsequent removal of the distal and proximal tendons of the soleus and the distal tendon and proximal gastrocnemius head as previously described[35].

For plantaris electroporation, a total of 30 µg of Myc-DDK tagged human METTL3 (Origene, RC200869) or a Myc-DDK empty vector was electroporated into the plantaris muscle of wild-type mice during synergist ablation surgery. In brief, on the morning of surgery, mice were quickly anesthetized and the lower hindlimb was injected with 30 µl of 2 mg/ml Hyaluronidase per limb (#P4D14907,

**Fig. 5 METTL3-mediated m⁶A modifications regulate the myostatin pathway. a** Schematic of the myofiber-specific Ribo-Tag WT and M3-mKO mice. **b** Western blot of HA expression and GAPDH control in muscles of Ribo-Tag expressing WT and M3-mKO mice or mice not expressing the tag (negative control; neg. ctrl). **c** Schematic of Ribo-seq protocol to capture muscle-specific ribosome-bound RNAs in WT and M3-mKO mice created in Adobe Illustrator. **d** Venn diagram showing the number of total enriched transcripts in myofiber ribosomes from METTL3 WT and mKO baseline muscle (big circles) and the number of differentially translated transcripts that also contain m⁶A at baseline (small circles). **e** Gene ontology (GO) analysis of Ribo-enriched, and m⁶A containing, as determined from baseline MeRIP-Seq samples. The enrichment score is based on a reference database of protein-coding genes; plotted GO categories have a false discovery rate (FDR) < 0.05. **f** Integrative Genomics View (IGV) of input and immunoprecipitation overlays on the *Acvr2a* gene from the MeRIP-seq baseline data set, and the Ribo-seq data sets for Ribo-Tag M3-mKO and WT. **g** Relative m⁶A enrichment, determined by qPCR analysis, following m⁶A immunoprecipitation in WT and M3-mKO plantaris muscles. **h** Relative ribosome occupancy enrichment, determined by qPCR analysis, of *Acvr2a* following Ribo-Tag immunoprecipitation in Ribo-Tag WT and M3-mKO plantaris muscles. **i** Immunofluorescence analysis of ACVR2A expression (red) in plantaris sections from WT and M3-mKO mice. **j** qPCR from RNA immunoprecipitation of *Acvr2a* mRNA in muscle using antibody against YTHDF1 (Y1), YTHDF2 (Y2), YTHDF3 (Y3), or normal IgG negative control (NC). **k** qPCR for *Acvr2a* post actinomycin treatment for the indicated times in 3T3 cells transfected with plasmids encoding for Myc-tagged Mettl3 (Myc-M3) or Myc alone control (Myc). Biological animal replicates: $n = 3$ per group in panel **d**–**f**, and **i**; $n = 4$ per group in panel **g** and **h**. Biological cell replicates: $n = 3$ per group in panels **j** and **k**. Data are presented as the mean ± SEM. *$P < 0.05$, by two-sided Student's *t* test for comparisons between WT and M3-mKO animals, or by two-way ANOVA with Tukey's HSD multiple-comparison test for comparison of the mean of WT and M3-mKO animal inputs and immunoprecipitations. Scale bar = 125 μm. Source data are provided as a Source Data file.

---

Worthington) with an insulin syringe (#309625, BD). Following 1 h of recovery, animals were re-anesthetized for surgery, and the soleus and gastrocnemius muscles were removed. Prior to suturing the animals, DNA solution was administered to the plantaris muscle, 2 sets of electrical shocks were applied (~180 V) and the hindlimb was then sutured and the mice placed in recovery cages. Animals returned to function movement within 15–30 min and tissues were harvested at experimental endpoint, 2 weeks following surgery.

Adeno-associated viral overexpression of METTL3 was achieved using AAV9 vectors produced at the Molecular Biotechnology Center at the University of Turin as previously described[36]. Neonatal mice were injected intraperitoneum with $1 \times 10^{12}$ viral genome particles of AAV9-METTL3 or control vectors, and analyzed 8 weeks later.

For myostatin inhibition the compound ACE-031 (soluble form of activin receptor type II, Direct-peptides, #ACE-031) was intraperitoneally injected at a dose of 10 mg/kg once a week (at the day of synergic ablation surgery and again 7-days later, over the course of a 14-day bilateral synergistic ablation).

The use of animals was approved by the Institutional Animal Care and Use Committee at The Ohio State University.

**Exercise metabolic testing.** Metabolic parameters were measured by indirect calorimetry. Mice were single housed in open-circuit OxyMax chambers as part of the Comprehensive Lab Animal Monitoring System (CLAMS) (Columbus Instruments) and analyzed using OxyMax software 2.4.2. For exercise tests, using previously described methods[37], mice were acclimated and then subjected to an endurance or graded maximal exercise test[37]. In brief, mice were placed on the treadmill at a 0° incline, and the shock grid was activated. The treadmill speed (meters), duration (minutes), and grade (degrees) were then increased until exhaustion as follows: 0 m/min, 3 min, 0°; 6 m/min, 2 min, 0°; 9 m/min, 2 min, 5°; 12 m/min, 2 min, 10°; 15 m/min, 2 min, 15°; 18, 21, 23, 24 m/min, 1 min, 15°; and +1 m/min, each minute thereafter. Exhaustion (endpoint for treadmill cessation) was defined as the point at which mice maintained continuous contact with the shock grid for 5 s. VO₂max was determined by the peak oxygen consumption reached during the test when the respiratory exchange ratio (RER) was greater than 1.0. The maximum running speed was defined as the treadmill speed at which VO₂max was achieved[37].

**Muscle force measurement.** In vivo plantar flexion torque measurements were performed using an in vivo muscle contractility apparatus (Model 1300A, Aurora Scientific Inc, Canada) as previously described[38]. In short, the right hind paw was taped to the force plate and positioned so that the foot and the tibia were aligned at 90°. The knee joint was securely clamped at the femoral condyles without compressing the nearby fibular nerve. Two disposable monopolar electrodes (Natus Neurology Inc, Middleton, WI, USA) were subcutaneously inserted over the tibial nerve for stimulation. Maximum plantar flexion tetanic torque (millinewton-meters [mN-m] was measured using a train of supramaximal 0.2 ms square-wave stimuli (150 Hz).

**Tissue staining and quantification.** Tissues were fixed in formalin, embedded in paraffin, and cut into 5 μm sections. For the determination of myofiber size, paraffin-embedded sections were deparaffinized, subjected to antigen retrieval for 15 min in boiling sodium citrate buffer (10 mM sodium citrate, pH 6.0, 0.05% Tween-20), rinsed in distilled water, incubated for one hour at room temperature with blocking buffer (3% goat serum, 0.05% Tween-20), and then incubated for 2 h at room temperature with Wheat Germ Agglutinin, Alexa Fluor™ 488 Conjugate (50 μg/ml, Invitrogen, #W11261). Immunostaining for ACVR2a was obtained on cryosections using antibodies from Santa Cruz Biotechnology (sc-515826; 1:250 dilution in 3% goat serum overnight at 4 °C). Slides were then rinsed and mounted

in Vectashield (Vector Labs, #H-1000-10) and imaged on the EVOS Imaging System (Invitrogen, Thermo Fisher Scientific). All images were imported into ImageJ 1.49v (National Institutes of Health, NIH) for quantification of myofiber size as previously described[39].

**Western blotting.** Protein extracts from whole skeletal muscles were generated using RIPA buffer (150 mM NaCl, 1% nonidet P-40, 0.5% sodium deoxycholate, 0.1% SDS, 25 mM Tris pH 7.4), centrifuged (4 °C × 12,000RPM × 20 min), and quantified using the Bio-Rad protein assay (Bio-Rad, # 5000001). Standard Western blotting analysis was performed using 10% SDS-PAGE gels with the following primary antibodies: METTL3 (1:2000, Abcam, #ab240595), GAPDH (1:10,000, Fitzgerald Industries, #10R-G109a), SMAD3 (1:1000, Cell Signaling, clone 67H9, #9523), Phospho-SMAD3 (1:1000, Cell Signaling, Ser423/425, clone 25A9, #9520), Puromycin (1:1000, EMD Millipore, clone 12D10, # MABE343), Phospho-AKT (1:1000, Cell Signaling #4060), AKT (1:1000, Cell Signaling #9272), Phospho-FOXO3 (1:1000, Cell Signaling #9466), and FOXO3 (1:1000, Cell Signaling #2497). Secondary antibody incubations were done at room temperature for 90 min using HRP-conjugated antibodies (1:10,000) and then imaged using a ChemiDoc system (Bio-Rad) as previously described[40].

**mRNA analysis by real-time PCR.** RNA was extracted using TRIzol (Thermo Fisher Scientific) and then reverse-transcribed using the High Capacity cDNA Reverse Transcription kit (Applied Biosystems) as previously described[41]. Selected gene expression differences were analyzed by real-time quantitative PCR (qPCR) using SsoAdvanced SYBR Green Supermix (Bio-Rad) in a 96-well format, and using Excel 16.16.24. Quantified mRNA levels were normalized to the house-keeping gene, and expression is presented relative to control levels. Primers used were: mouse *Actb* 5′-TGTGATGGTGGGAATGGGTCAGAA-3′ and 5′-TGTGGT GCCAGATCTTCTCCATGT-3′; mouse *Mettl3* 5′-GTGCATGAAAGCCAGTGA CG-3′ and 5′-CTTGCTGCCAGGACTCTCAG-3′; human *METTL3* 5′-TCAAGG AAACATGCTGCCTCA-3′ and 5′-ACAGGGTCGATCAGCATCAC-3′; mouse *Murf1* 5′-FGCTGGTGGAAAA CATCATTGACAT-3′ and 5′-RCATCGGGTGGC TGCCTTT-3′; mouse *Mafbx* 5′-FCTTTCAACAGACTGGACTTCTCGA-3′ and 5′-RCAGCTCCAACAGCCTTACTACGT-3′; and mouse *Acvr2a* 5′-FGCAAGGT TGTTGGCTGGATG-3′ and 5′-RTGGGCTGTGTGACTTCCATC-3′.

**RNA immunoprecipitation and decay.** For RNA immunoprecipitation from mouse muscles, the tissue was excised and then chopped and submerged in RNA Later (ThermoFisher; AM7020). Tissues were fixed in 0.15% formaldehyde, quenched in 1.25 M glycine for 5 min at room temperature, washed with ice-cold PBS, and sonicated in Buffer A (100 mM KCL, 5 mM MgCl2, 10 mM Hepes pH 7.0, 0.5% NP-40, 1 mM Dithiothrectol (DTT), protease inhibitors). Sonicated samples were clarified by centrifugation at 4 °C for 15 min. Pull-down was performed by incubating 1 mg of protein extract with 5 μg of antibody (anti-YTHDF1, #17479-AP, ProteinTech; anti-YTHDF2, #ab220163, Abcam; anti-YTHDF3, #sc-377119, Santa Cruz Biotechnology; anti-mouse IgG, #sc-2025, Santa Cruz Biotechnology; anti-rabbit IgG, #12-370, EMD Millipore) at 4 °C for 4 h, followed by the addition of 40 μL of protein A/G magnetic beads (#88803, Pierce) to the rotation overnight. The following day samples were washed 5 times in Buffer B (50 mM Tris pH 7.4, 150 mM NaCl, 1 mM MgCl2, 0.05% NP-40), subjected to DNase (#AM2238, Invitrogen) digest at 37 °C for 5 min, and then eluted in Buffer B, supplemented with 1 mg/ml Proteinase K (#E195-5ML, VWR) and 0.1% SDS, at 55 °C for 30 min. Following elution, RNA from input and immunoprecipitation samples was isolated using standard phenol-chloroform extraction followed by reverse transcription and qPCR as previously described[41]. mRNA decay experiments were performed in

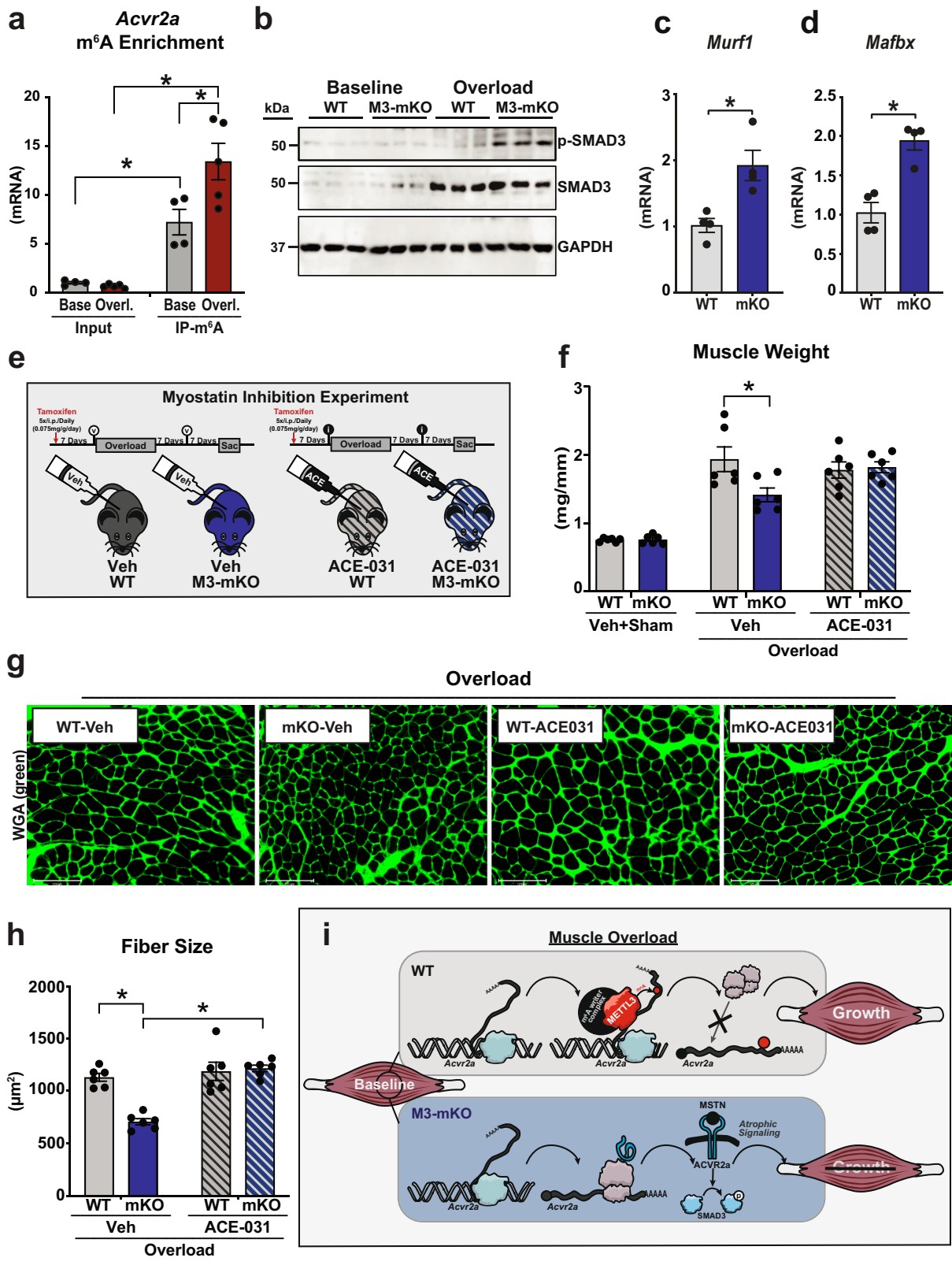

3T3-L1 murine cell lines expressing the Myc-tagged METTL3 or a Myc empty vector control. mRNA decay experiments were performed 48 h after transient transfection with Lipofectamine 3000 (# L3000008, ThermoFisher) using 5 μg/mL Actinomycin D (#A1410, Sigma) as a transcription inhibitor.

**m⁶A quantification, immunoprecipitation, and sequencing**. RNA was extracted from mouse skeletal muscles using Trizol (Life Technologies) in baseline or synergist

ablation overloaded plantaris muscles and subjected to m⁶A quantification using the m⁶A RNA Methylation Quantification Kit (Colorimetric) (#ab185912, Abcam) in biological triplicate as previously described[42]. For genome-wide m⁶A profiling, total RNA was isolated from muscle plantaris of baseline and overloaded WT mice using Qiagen RNeasy Fibrous Tissue Mini Kit (# 74704, Qiagen) per manufacturer's instructions. Total RNA samples were pooled to a total RNA amount of ~20 mcg/sample and processed using Invitrogen PolyA+ RNA selection (Dynabeads mRNA Purification Kit (Cat # 61006) according to the manufacturer's protocol. RNA was

**Fig. 6 Hypertrophy defects in METTL3-mKO mice are rescued with myostatin inhibition. a** Relative m⁶A enrichment, determined by qPCR analysis, following m⁶A immunoprecipitation in WT plantaris muscles following 7 days of sham (base) or overload (overl) surgeries. **b** Western blot of p-SMAD3 (p = phospho), total SMAD3 and GAPDH expression in muscles of WT and M3-mKO mice at baseline or 7 days following muscle overload. **c** qPCR analysis of *Murf1* (muscle-specific ring finger protein 1) and **d** *Mafbx* (muscle-atrophy F-box protein) expression in WT and M3-mKO muscles. **e** Schematic of myostatin inhibition experimental plan created in Adobe Illustrator. **f** Plantaris weight, **g** representative wheat germ agglutinin (WGA; green) stained images, and **h** fiber size in day 14 overloaded WT and M3-mKO muscles treated with and without myostatin inhibitor ACE-031. **i** Descriptive figure of working model created in Adobe Illustrator. As muscle undergoes a hypertrophic response to overload, the METTL3 complex allows for the distribution of m⁶A to the *Acvr2a* transcript, which prevents its translation, and allows for normal muscle growth. In M3-mKO animals, there is no METTL3 to distribute m⁶A on the *Acvr2a* transcript, thus promoting *Acvr2a* translation, myostatin (MSTN) activity, and atrophic signaling through SMAD3 phosphorylation (p). Biological animal replicates: $n = 4$ (baseline) and 5 (overload) in panel **a**; $n = 4$ per group in panel **c** and **d**; and $n = 4$ per group in panel **f** and **h**. Data are presented as the mean ± SEM. *$P < 0.05$, by two-sided Student's *t* test for comparisons between WT and M3-mKO animals, or by two-way ANOVA with Tukey's HSD multiple-comparison test for comparison of the mean of WT and M3-mKO overloaded animal at with and without inhibitor ACE-031. Scale bar = 125 μm. Source data are provided as a Source Data file.

then fragmented using NEBNext Magnesium RNA Fragmentation Buffer (#6150 S, NEB) for 10 min at 94 °C, ethanol precipitated overnight, and resuspended in water. Enrichment of N6-methyladenylated RNA by immunoprecipitation was performed using the NEB enrichment kit according to manufacturer's instructions (#E1610S, NEB) with 100 ng of fragmented mRNA being retained for sample inputs. In short magnetic A/G beads (#88802, Thermofisher Scientific) were prepared, washed, and 1 μl of N6-Methyladenosine Antibody/per sample was added to magnetic beads. Following wash-steps, 1 μg of fragmented mRNA was added to re-suspended antibody bound beats and subjected to rotation for 1 h at 4 °C. Beads were then washed with a serious of low to high salt buffers and then eluted using Buffer RLT (#79216, Qiagen). NGS Illumina sequencing library preparation on input and immunoprecipitation samples was performed using the Takara picov2 sequencing kit (#634878, Takara) according to manufacturer instructions, following RNA cleanup by ethanol precipitation and quality-control check by bioanalyzer. Prepared libraries were submitted for RNA sequencing at the University of Chicago Genomics Facility on an Illumina HiSeq 2000 machine in single-end read mode with 50 bp per read.

**Ribosome immunoprecipitations and sequencing of translating mRNAs.** Plantaris muscles were excised from WT and M3-mKO myofiber-specific mice, immediately micro-punctured, and washed in 100 μg/ml cycloheximide (CHX) on ice. Muscles were then nap frozen and liquid nitrogen pulverized (#40355, Cole Parmer) into coarse powder and then incubated with rotation at 4 °C for 15 min in polysome isolation buffer (20 mM Tris, 10 mM MgCl2, 200 mM KCl, 2 mM DTT, 1% TX, 100 μg/mL CHX). Samples were clarified at $10,000 \times g \times 20$ min × 4 °C and protein concentrations were determined. In all, 1/10 of clarified lysates were set aside as inputs and 1 mg/mL protein was utilized for immunoprecipitation using 4 μg/mL of anti-HA antibody (HA.11, Covance) and rotated at 4 °C for 4 h. Following incubation with of the lysate and antibody, magnetic A/G beads were added and left to mix at 4 °C with rotation overnight. The following day, beads were washed three times in high salt buffer and eluted by the addition of 300 μl of Trizol and vortex. Trizol containing input and immunoprecipitation samples were then subjected to Trizol extraction as described above and shipped for library preparation (DNBseq Eukaryotic -T resequencing, transcriptome library, BGI Americas) and RNA sequencing (Complete Genomics, BGI Americas).

**Bioinformatics analyses.** Data preparation for sequencing analysis was performed as previously described[43]. In short, Fastq files were checked for quality using FASTQC (https://www.bioinformatics.babraham.ac.uk/projects/fastqc/). Processed fastq files were aligned using Hisat2[44] against the Ensembl index for mice (Genome Reference Consortium mouse mm10, Release 21, GRCm38.p6) and sorted using Samtools[45]. Sorted BAM files from Hisat2 were used as input for calculating forming count matrices using HTSeq[46]. GTF files matched to their respective Ensembl alignment indexes were used[47]. Sorted BAM files from Hisat2 were further used as input for exomePeak2 analysis (https://github.com/ZW-xjtlu/exomePeak2). This includes peak enrichment of both meRIP-seq analysis and Ribo-seq analysis. GTF files for exomePeak2 matched their respective Ensembl alignment indexes. Peak prediction used general default settings for exomePeak2, but in short, minimal peak length was set to 25 bp, peak cutoff FDR was set to 0.05, and fold enrichment was set to 4. When looking for enriched peaks between sample groups, fold enrichment was set to 1.5. Output bed files from exomePeak2 analysis of meRIP-seq data were used as input for global peak distribution analysis (RNA features, biotypes, density plots, etc.) using RNAmod, an integrated system for the annotation of mRNA modifications[48]. Analysis was done with general default settings. Sequencing data have been deposited in the NIH Gene Expression Omnibus under accession code GSE179368. Gene ontology analysis on detected and enriched peaks from exomePeak2 data were carried out using the WEB-based Gene set analysis toolkit (Webgestalt)[49].

**Statistical analysis.** All results were presented as mean ± SEM, with dots indicating individual biological samples within a group. Each experiment was repeated a minimum of three independent times to ensure reproducibility. Statistical analysis between two groups was performed using the Student's two-tailed *t* test for normally distributed data with a p-value of ≤0.05 considered significant. For groups of 3, a one-way ANOVA followed by Tukey's honestly significant difference (HSD) multiple-comparison test was performed, with statistical significance set at $\alpha = 0.05$. For groups of 2 genotypes and 2 conditions, a two-way ANOVA followed by Tukey's HSD multiple-comparison test was performed, with statistical significance set at $\alpha = 0.05$. Bioinformatic statistics used default settings for each individual pipeline. In short, DESeq2 used a Benjamini and Hochberg adjusted p-value. ExomePeak2 used both FDR calculation and rescaled hypergeometric test to calculate predicted and differentially enriched peaks. Data analysis was performed using GraphPad Prism 8 (GraphPad Software).

**Reporting summary.** Further information on research design is available in the Nature Research Reporting Summary linked to this article.

## Data availability

Source Data are provided with this paper. Sequencing and bioinformatics analysis are available through Gene Expression Omnibus submission GSE179368. All other data generated and/or analyzed during the current study are available from the correspondent author on reasonable request. Source data are provided with this paper.

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

## Acknowledgements

This work was supported by the NIH under grants [R01 HL 136951 and R01 HL 154001] to F.A., F31 AR073638 to J.M.P, RM1 HG008935 to C.H., and by the US-Israel Binational Science Foundation (BSF) grant number 2017094.

## Author contributions

J.M.P. and F.A. conceived the project. J.M.P. and F.A. wrote and revised the manuscript. J.M.P. performed most of the in vivo and ex vivo studies involving: animal surgeries, animal electroporation, animal injections, animal exercise testing, tissue harvesting and collection, protein and RNA extractions, western blotting, real-time PCR, histological analysis (embedding, cutting, sectioning, staining, imaging, and quantification), m6A immunoprecipitations, RNA and Ribotag immunoprecipitations. W.D.A. and C.C.I. performed in vivo torque testing. L.D. performed m6A ELISAs. V.A.G. optimized RNA immunoprecipitations. J.M.P. and J.M.B. performed MeRIP-Seq and library preparation. H.L.S. and C.H. performed MeRIP-Seq sequencing. J.B. and S.A.H. performed MeRIP-Seq analysis. S.A.H. performed RiboTag sequencing analysis.

## Competing interests

C.H. is a scientific founder and a member of the scientific advisory board of Accent Therapeutics, Inc. The remaining authors declare no competing interests.
