## [Peer Review File · Nature Communications]

The m⁶A methyltransferase METTL3 regulates muscle maintenance and growth in miceREVIEWER COMMENTS

Reviewer #1 (Remarks to the Author):

In this study Petrosino et al uncovered a novel role for the m6A mRNA modification in muscle maintenance and growth. This modification is the most prevalent on mRNA and was shown to affect a wide range of physiological processes by altering the RNA fate. Here the authors investigated its potential role in muscle physiology. They found that the m6A level increases upon muscle overload induced hypertrophy and identified the transcripts with methylation changes. Gain and loss of function of Mettl3 revealed a critical function of m6A in muscle growth, both upon acute and chronic treatments. Lastly the authors found that the translation of activin receptor was increased upon Mettl3 loss, and that inhibiting its activity rescued muscle weight and fiber size of Mettl3 KO mice, suggesting that this a critical target of Mettl3 in this process.

This is an interesting piece of work that highlights a novel function for m6A in muscle growth, which can be of interest for the RNA community as well as for future studies aiming at slowing down the loss of muscle strength and size that occur upon aging. The manuscript is well written and easy to read. Nevertheless, the mechanistic part is rather weak, in particular the conclusion that Acvr2a is a direct target would need more support.

1. The effect of m6A on Acvr2a is somewhat unexpected. m6A is best known for stimulating decay and translation of its targets. Here the authors suggest that m6A inhibits the translation of Acvr2a. How does this happen? Is the m6A peak localized near stop codon (it's difficult to see in panel 5f). The authors suggest that Ythdf2 could be involved. What is the evidence that Ythdf2 inhibits translation? If it's an effect on mRNA decay the authors should provide also the evidence for this.

2. Figure 5d is unclear and is not described in the figure legend. What is it shown exactly? Which m6A containing transcripts are meant (m6A in the baseline or overload condition)? What is the overlap of the Ribo enriched transcripts in the WT and Mettl3 KO conditions? How many were identified in total? What are the transcripts that did not change? That only 28% of differentially translated mRNAs in Mettl3 KO have m6A seems surprisingly low (considering that 3000 transcripts were identified). What is the overlap with the transcripts that show an altered m6A level in the overload condition?

3. Figure 5f: The meRIP was done in the baseline and overload condition, however, it is just shown that Acvr2a is methylated in the baseline condition. How does it change in the overload condition? Does the m6A level increase? (according to the suggested model it should be)

4. Figure 5h is just showing the ribosomal occupancy. It is not shown that the protein level of the receptor is also higher. Is the translation actually dependent on the m6A site or could it be an indirect effect? What about other activin receptors?

5. Figure 6: Does the inhibition of the activin receptors increase the effect of Mettl3 overexpression?

6. The effect seen on the activin pathway is interesting. The authors could discuss further these findings in light of the current literature. In particular SMAD transcription factors were proposed to direct the methyltransferase complex to its targets, thereby enhancing m6A deposition. Perhaps there is a negative feedback mechanism that attenuates Acvr2 translation after increased m6A.

Conclusion:

On its current form the conclusion appears too strong. Even though the data definitively suggest a relationship between m6A/Mettl3 and muscle development, direct mechanistic evidence is not provided. The part about the translational effect can be definitely improved. I feel that the authors did not fully take advantage of their Ribo-Tag data. Also the fact that translation is inhibited is unusual, which would deserve a better understanding (e.g which is the reader involved?)

Reviewer #2 (Remarks to the Author):

The study of Pertosino evidencing the role of translation regulation of muscle growth through the methylation of adenosine by the enzyme METTL3 is certainly a new avenue of research into our understanding of post-natal muscle growth.

The main case of the study posits that methylation of adenosine targets the activin IIA receptor which leads to its attenuated translation resulting in a diminution of signalling that would normally inhibit protein synthesis and activate catabolic pathways.

The study is certainly interesting. I would like to offer a number of avenues that the authors should investigate to strengthen their study with the ultimate aim of establishing the robustness of their conclusions.

(1) As the authors will know, Activin and Myostatin signal through the ActRIIa and RIIb. The authors need to investigate in detail the response of the ActRIIb in their interventions, with the aim of addressing whether there is functional redundancy. There is a huge volume of data that shows that most of the muscle related signalling occurs through the actRIIB receptor.

(2) What explanation do the authors have from a molecular perspective that only the ActRIIA is targeted and not the ActRIIB.

(3) The role of METTL3 in promoting muscle mass in the absence of overload was investigated in P3 mice. However there is still on going stem cell accretion to muscle fibres at this stage.

Therefore the changes in muscle mass will be through a number of mechanisms. These experiments need to be conducted in adult mice (older than 4 months).

(4) The loss muscle in the chronic deletion experiments (fig 4) are very modest. I wonder whether some sort of compensation has occurred in this time period. The authors need to characterize this experiment in a shorter time frame. It is essential again to evaluate that the protein level the amount of ActRIIA and ActRIIB.

(5) It is essential that given the authors propose that METTL3 is ultimately regulating both anabolic and catabolic pathways the protein markers be assessed. To that end they need to examine levels of activated Akt and downstream polypeptides as well as the activations of drivers of UE ligases including the FoxO family.

(6) Was there any influence of sex of mice and the changes in levels of METTL3 in muscle?

Reviewer #1 (Remarks to the Author):

In this study Petrosino et al uncovered a novel role for the m6A mRNA modification in muscle maintenance and growth. This modification is the most prevalent on mRNA and was shown to affect a wide range of physiological processes by altering the RNA fate. Here the authors investigated its potential role in muscle physiology. They found that the m6A level increases upon muscle overload induced hypertrophy and identified the transcripts with methylation changes. Gain and loss of function of Mettl3 revealed a critical function of m6A in muscle growth, both upon acute and chronic treatments. Lastly the authors found that the translation of activin receptor was increased upon Mettl3 loss, and that inhibiting its activity rescued muscle weight and fiber size of Mettl3 KO mice, suggesting that this a critical target of Mettl3 in this process.

This is an interesting piece of work that highlights a novel function for m6A in muscle growth, which can be of interest for the RNA community as well as for future studies aiming at slowing down the loss of muscle strength and size that occur upon aging. The manuscript is well written and easy to read. Nevertheless, the mechanistic part is rather weak, in particular the conclusion that Acvr2a is a direct target would need more support.

We thank the reviewer for their work in assessing our manuscript and helping us improve it.

1. The effect of m6A on Acvr2a is somewhat unexpected. m6A is best known for stimulating decay and translation of its targets. Here the authors suggest that m6A inhibits the translation of Acvr2a. How does this happen? Is the m6A peak localized near stop codon (it's difficult to see in panel 5f). The authors suggest that Ythdf2 could be involved. What is the evidence that Ythdf2 inhibits translation? If it's an effect on mRNA decay the authors should provide also the evidence for this.

We appreciate this reviewer's comment. We have now performed further experimentation that proves that m⁶A affects Acvr2a mRNA stability. Indeed, METTL3 accelerated Acvr2a decay in a time course post transcription inhibition (**new Figure 5k**). We also tested which m⁶A binding protein of the YTHDF family acts on Acvr2a mRNA and found that YTHDF2 is its predominant interactor (**new Figure 5j**). We have accordingly modified the discussion to reflect these new results (**page 9** of the discussion section). Additionally, we better visualize the m⁶A peak in panel 5f to clarify the modification is located near the stop codon (**Figure 5f**).

2. Figure 5d is unclear and is not described in the figure legend. What is it shown exactly? Which m6A containing transcripts are meant (m6A in the baseline or overload condition)? What is the overlap of the Ribo enriched transcripts in the WT and Mettl3 KO conditions? How many were identified in total? What are the transcripts that did not change? That only 28% of differentially translated mRNAs in Mettl3 KO have m6A seems surprisingly low (considering that 3000 transcripts were identified). What is the overlap with the transcripts that show an altered m6A level in the overload condition?

We apologize for the lack of clarity on figure 5d. We have now updated the correspondent legend. Because the Ribo-tag data was generated at baseline, we have focused the overlap analysis on the baseline m⁶A profile. Manipulation of METTL3 level drove a muscle phenotype over time even in the absence of the overload stress; we therefore reasoned the baseline condition is important to explain the

most general mechanism of action of this pathway in muscle. Following this reviewer's suggestion we did perform an overlap analysis using m⁶A data from muscle overload. However, the overlap was not very pronounced and we think this is not surprising since the muscle transcriptome profoundly changes during overload. With this in mind we reasoned comparing transcripts originated from the same baseline condition is more informative. We now better explain our results and specify that we focused on the baseline condition in both result and legend sections. We also added that out of 557 differentially translated transcripts (453 enriched in WT, plus 104 enriched in mKO), 216 were m⁶A containing (178 in the WT enrichment, plus 38 in the mKO enrichment) (**page 7** of the result section). We also corrected that the percentage of m⁶A-containing transcripts that are differentially ribo-enriched is actually 39% instead of 28 (**page 7** of the result section). Importantly, because the bioinformatics analysis shows differentially translated transcripts (from the mKO versus WT comparison an originated positive integer represents an enrichment in mKO, while a negative integer represents an enrichment in WT), we can consider that all other transcripts expressed in muscle would be unchanged between WT and mKO. We apologize for our previous imprecision and the confusion that this has created.

3. Figure 5f: The meRIP was done in the baseline and overload condition, however, it is just shown that Acvr2a is methylated in the baseline condition. How does it change in the overload condition? Does the m⁶A level increase? (according to the suggested model it should be)

We have now included analysis of Acvr2a methylation comparing baseline and overload condition. According to the model, we saw increased m⁶A level on Acvr2a mRNA following overload (**new Figure 6a**). We thank the reviewer for suggesting this addition as we agree with the reviewer that this new data strengthened our conclusion.

4. Figure 5h is just showing the ribosomal occupancy. It is not shown that the protein level of the receptor is also higher. Is the translation actually dependent on the m⁶A site or could it be an indirect effect? What about other activin receptors?

The reviewer raised an important point. We have now included protein analysis for Acvr2a receptor showing the increased level of this receptor on the plasma membrane of METTL3-KO muscle (**new Figure 5i**). While all of our analyses point at a direct link between Acvr2a m⁶A modification and its translation, we recognize that we cannot currently exclude the contribution of indirect effects. We now discuss this limitation in **page 9** of the discussion. We also analyzed a potential m⁶A role on the other major activin receptor, Acvr2b. However, the results on this transcript show strong downregulation of this transcript level with overload more than a role for METTL3-m⁶A (**new Supplementary Figure 3a-c**). This is interesting as it could suggest transcriptional regulation of Acvr2b, as opposed to post-transcriptional control of Acvr2a expression. We now discuss this data in **page 9** of the discussion.

5. Figure 6: Does the inhibition of the activin receptors increase the effect of Mettl3 overexpression?

This is an interesting point. METTL3 overexpression is sufficient to enhance the hypertrophic response of muscle. Our working model suggests that inhibition of Acvr2a by METTL3 is important for this effect. We can speculate that further muscle growth could be stimulated by addition of inhibitors that would globally affect the activin receptors. However, there are constraints to how much a muscle actually grows in a stimulated animal, and a plateau is typically observed *in vivo*. With this in mind, it would be hard to interpret the results obtained from an experiment with simultaneous METTL3 overexpression and

inhibition of activin receptors: no additive effect could suggest that METTL3 is sufficient to efficiently inhibit activin receptor signaling; however, we would not be able to exclude that maximal growth potential was already reached with METTL3 overexpression alone. Because of this limitation we focused instead on the growth rescue of METTL3 deficiency by activin receptor inhibition.

6. The effect seen on the activin pathway is interesting. The authors could discuss further these findings in light of the current literature. In particular SMAD transcription factors were proposed to direct the methyltransferase complex to its targets, thereby enhancing m6A deposition. Perhaps there is a negative feedback mechanism that attenuates Acvr2 translation after increased m6A.

This is a great suggestion. Previous literature showing an interaction between SMADs and the m⁶A methyltransferase complex further supports a mechanistic intersection between the two pathways. We now discuss this (**page 9** of the discussion) and cite PMID 29489750 in support of this concept.

Conclusion:

On its current form the conclusion appears too strong. Even though the data definitively suggest a relationship between m6A/Mettl3 and muscle development, direct mechanistic evidence is not provided. The part about the translational effect can be definitely improved. I feel that the authors did not fully take advantage of their Ribo-Tag data. Also the fact that translation is inhibited is unusual, which would deserve a better understanding (e.g which is the reader involved?)

We thank this reviewer for the great help provided. We have taken into accounts all suggestions and accordingly reinforced the study. Related to the key points summarized in the conclusion, we now better explain the Ribo-tag data, we show that METTL3 affects mRNA decay and we provide evidence that YTHDF2 is the predominant reader interacting with Acvr2a transcript. We appreciate the opportunity to revise the manuscript and we hope we satisfactorily clarified this reviewer's concerns.

Reviewer #2 (Remarks to the Author):

The study of Pertosino evidencing the role of translation regulation of muscle growth through the methylation of adenosine by the enzyme METTL3 is certainly a new avenue of research into our understanding of post-natal muscle growth.

The main case of the study posits that methylation of adenosine targets the activin IIA receptor which leads to its attenuated translation resulting in a diminution of signalling that would normally inhibit protein synthesis and activate catabolic pathways.

The study is certainly interesting. I would like to offer a number of avenues that the authors should investigate to strengthen their study with the ultimate aim of establishing the robustness of their conclusions.

We thank this reviewer for their help strengthening our study.

(1) As the authors will know, Activin and Myostatin signal through the ActRIIa and RIIb. The authors need to investigate in detail the response of the ActRIIb in their interventions, with the

aim of addressing whether there is functional redundancy. There is a huge volume of data that shows that most of the muscle related signalling occurs through the actRIIB receptor.
(2) What explanation do the authors have from a molecular perspective that only the ActRIIA is targeted and not the ActRIIB.

We agree with this reviewer and have now provided complementary analysis on Acvr2b transcript. We found that the strongest level of regulation of Acvr2b occurs through downregulation of its mRNA content following overload stimulation (**new Supplemental Figure 3a**), while the impact of m⁶A and METTL3 was less clear (**new Supplemental Figure 3b,c**). This would suggest transcriptional control of this messenger to complement the m⁶A-dependent post-transcriptional regulation of Acvr2a. This is interesting as it indicates a diversification of gene expression regulation on the Acvr2 family of transcripts in muscle. Considering the major importance of activin receptor signaling in muscle, we can speculate on the evolutionary advantage of such diverse modulation. This is now discussed in **page 9** of the revised discussion.

(3) The role of METTL3 in promoting muscle mass in the absence of overload was investigated in P3 mice. However there is still on going stem cell accretion to muscle fibres at this stage. Therefore the changes in muscle mass will be through a number of mechanisms. These experiments need to be conducted in adult mice (older than 4 months).

The reviewer raised an important point. While overload-induced hypertrophy in adult muscle and postnatal muscle growth are distinct processes, they both involve growth stimuli. Following this reviewer's suggestion we performed a METTL3 overexpression experiment by injecting AAV-METTL3 or control vectors into the tibialis anterior muscle of 5 month old mice and sacrificed the animals 8 weeks later. The result shows a strong trend toward increased muscle mass with higher METTL3 level in adult muscle, and a significant increase in myofiber size (**new Figure 3k-n**). Overall, the results suggest that in addition to operating in the context of growth stimuli, METTL3 can also play a role in promoting muscle mass in the absence of other triggers.

(4) The loss muscle in the chronic deletion experiments (fig 4) are very modest. I wonder whether some sort of compensation has occurred in this time period. He authors need to characterize this experiment in a shorter time frame. It is essential again to evaluate that the protein level the amount of ActRIIA and ActRIIB.

This is another interesting point. We had an ongoing cohort of mice where the deletion was initiated about 6 months prior to receiving the reviewers' comment and we sacrificed those mice immediately after reading this comment to have an additional time frame for the chronic deletion experiment. With 6 months of METTL3 deletion the loss of muscle mass was already significant or close to significance for most muscle types, although the phenotype was not stronger than observed with 8 months of deletion (**new Supplementary Figure 2a-h**). We also performed a deletion experiment where we assessed the animals 5 weeks later. This time point did not show any significance effect of METTL3 deletion on muscle mass, suggesting more time is needed for phenotype onset (**new Supplementary Figure 2i-p**). This is interesting as it indicates a progressive wasting phenotype that could be due to changes in the activin receptor ligands as the mouse ages. We have also confirmed increased ActRIIA expression in METTL3-deficient muscle (**new Figure 5i**), while ActRIIB did not seem to follow an m⁶A-dependent regulation, as detailed in our above response to comments #1 and 2. We expanded our discussion section to incorporate the new information gained by these experiments (**page 9 and 10**).

(5) It is essential that given the authors propose that METTL3 is ultimately regulating both anabolic and catabolic pathways the protein markers be assessed. To that end they need to examine levels of activated Akt and downstream polypeptides as well as the activations of drives of UE ligases including the FoxO family.

This is yet another good point. We are now including western blots showing the phosphorylation status of Akt and FoxO3 (**new Supplementary Figure 4a,b**). Interestingly, although regulation of these pathways was generally not as dramatic as the one shown for SMAD3, lower levels of phospho-Akt were observed in overloaded muscle from METTL3 mKO mice. A similar trend was observed for the Akt downstream target FoxO3 (which phosphorylation by Akt inhibits its function) although some variability in this regulation was observed. While we also checked FoxO1, we did not see any signal at the expected molecular weight in our samples, suggesting its activation might not be occurring in the tested experimental conditions. Overall, the reviewer is right that other pathways than SMAD are likely contributing to a dysregulated balance between anabolic and catabolic events orchestrated by METTL3.

(6) Was there any influence of sex of mice and the changes in levels of METTL3 in muscle?

We did not see sex differences in the reported METTL3-dependent effects or METTL3 levels themselves; we therefore utilized both sexes in our experiments.

We are extremely grateful to this reviewer for their help and we believe that thanks to their suggestions the paper is now stronger.

REVIEWERS' COMMENTS

Reviewer #1 (Remarks to the Author):

The authors addressed all of my points, performed the necessary additional analysis, and updated the manuscript accordingly. I am thus in favor of publication.

Reviewer #2 (Remarks to the Author):

The authors have addressed all the points that I raised to with extra experimentation.

Reviewer #1 (Remarks to the Author):

The authors addressed all of my points, performed the necessary additional analysis, and updated the manuscript accordingly. I am thus in favor of publication.

Reviewer #2 (Remarks to the Author):

The authors have addressed all the points that I raised to with extra experimentation.

We are extremely grateful to the reviewers for their help improving our paper and their positive reception of the revised study.